# Finger Millet Production in Ethiopia: Opportunities, Problem Diagnosis, Key Challenges and Recommendations for Breeding

Adane Gebreyohannes [1,2,*], Hussein Shimelis [1], Mark Laing [1], Isack Mathew [1], Damaris A. Odeny [3] and Henry Ojulong [3]

1   African Centre for Crop Improvement, School of Agricultural, Earth and Environmental Sciences, University of KwaZulu-Natal, Scottsville 3209, South Africa; shimelish@ukzn.ac.za (H.S.); laing@ukzn.ac.za (M.L.); isackmathew@gmail.com (I.M.)
2   Ethiopian Institute of Agricultural Research, Melkassa Agricultural Research Center, Adama 436, Ethiopia
3   International Crop Research Institute for Semi-Arid Tropics, Nairobi P.O. Box 39063-00623, Kenya; d.odeny@cgiar.org (D.A.O.); h.ojulong@cgiar.org (H.O.)
*   Correspondence: adanegy10@gmail.com

**Abstract:** Finger millet (*Eleusine coracana* (L.) Gaertn) is a highly nutritious crop, predominantly grown in the semi-arid tropics of the world. Finger millet has a niche market opportunity due to its human health benefits and being rich in calcium, iron and dietary fiber and gluten-free. Ethiopia is the center of the genetic diversity of the crop. However, the productivity of finger millet in the country is low (<2.4 tons ha$^{-1}$) compared with its potential yield (6 tons ha$^{-1}$). The yield gap in Ethiopia is due to a range of biotic and abiotic stresses and socio-economic constraints that are yet to be systemically documented and prioritized to guide future production and improved variety development and release. The objective of this study was to document finger millet production opportunities, constraints and farmer-preferred traits in Ethiopia as a guide to variety design in improvement programs. A participatory rural appraisal (PRA) study was undertaken in six selected districts of the Southern Nation Nationalities People Region (SNNPR) and Oromia Region in Ethiopia. Data were collected from 240 and 180 participant farmers through a semi-structured questionnaire and focus group discussion, respectively. Finger millet was the most important crop in the study areas grown mainly for a combination of uses, including for food, feed and cash (reported by 38.8% of respondent farmers), food and feed (14.5%), food and cash (13.7%), food (11.5%) and food, cash, feed and construction material (9.7%). Hand weeding was used by 59.2% of the respondent farmers, followed by hand weeding and chemical herbicides (40.8%). Finger millet was mainly planted as a sole crop (reported by 97.0% respondents), mixed (1.7%) and sole and mixed (1.3%). About 75.6% of respondent farmers only practiced finger millet rotation with other crops. Respondent farmers indicated their source of fresh seed was from the Bureau of Agriculture (49.1%), farmer-to-farmer seed exchange (22.1%), own saved seed (7.5%), local producers (7.5%), research institutions (5.8%), unknown sources (4.1%), local market (3.5%) and cooperatives (0.42%). The total cost of finger millet production per hectare was calculated at 1249 USD with a total income of 2139 USD/ha, making a benefit to cost ratio of 1.71:1.00 and indicating the relatively low yield gains using the currently grown varieties. The main constraints to finger millet production in the study areas were drought stress (reported by 41.3% respondents), lack of improved varieties (12.9%), lack of financial resources (11.3%), small land holdings (10.8%), limited access to seed (10.0%), a shortage of fertilizers (5.4%), poor soil fertility (4.6%), shortage of draught power (1.3%), labour shortages (1.3%) and high labour costs (1.3%). The most important farmer-preferred traits in a finger millet variety were high grain yield, compact head shape, 'enjera'-making quality, high marketability and early maturity, resolved through principal component analysis. The above-mentioned production constraints and farmer-preferred traits are strategic drivers to enhance finger millet productivity and need to be incorporated into Ethiopia's finger millet breeding and technology development.

**Keywords:** *Eleusine coracana*; drought stress; finger millet; participatory rural appraisal; indigenous knowledge

## 1. Introduction

Finger millet (*Eleusine coracana* (L.) Gaertn) is an important cereal crop in the semi-arid and tropical regions of the world. The name finger millet is derived from the appearance of spikes or fingers, which are arranged and appear like human fingers. Compared with other major cereals such as rice, wheat and barley, it is relatively drought-tolerant due to its C4 photosynthesis system and adaption to grow under harsh and marginal agro-ecologies. Agriculture is an important economic sector in Africa, including Ethiopia. The sector accounts for 25% of Africa's GDP, 21% of exports, and 65–70% of the workforce supporting the livelihoods of 90% of population [1–3]. In Ethiopia, agriculture contributes to 44% of GDP, 70% of export earnings and 80% of employment opportunity [4]. Finger millet is grown mainly for its grain, which is utilized to make traditional food and drinks, while the stalks are used for livestock feed, construction and fuel. Finger millet has various human health benefits such as reducing diabetes [5], obesity [6], osteoporosis [7,8], anemia [6], malaria [9,10] and diarrhea [9,10]. The health values of finger millet are linked to its high calcium, iron and dietary fiber content and being gluten-free. These health benefits will render finger millet as a crop of niche market opportunity in the future. Finger millet is cultivated in more than 25 countries in Africa and Asia [11]. Ethiopia is the second largest producer of finger millet in the world after India [12,13]. In Ethiopia, the grain is processed to make unleavened bread (locally referred to as enjera) and for malting to prepare local drinks such as a distilled spirit 'Areki' or local beers such as 'tella' and non-alcoholic drinks such as 'karibu' and 'shamita', while the straw is vital as a livestock feed and for thatching of houses [14,15].

The global production area and total production for finger millet are unknown since both statistics are merged and reported with other millets. An estimated total production area of 32,554,127 ha is devoted to millets production worldwide [12]. It is estimated that the share of the global finger millet production area is about 12.5% of the millet. Ethiopia's total finger millet production area is 455,581 ha [16], making an 11.2% global share [12,16]. A total of 3,834,021 tons of finger millet grain is produced per annum globally [12], while Ethiopia's output is estimated at 1,125,958 tons [16], equivalent to 29.4% of global production. Finger millet is the sixth most important cereal crop in Ethiopia in total area and production after tef (*Eragrostis tef* (Zucc.)Trotter), maize (*Zea mays* (L.)), sorghum (*Sorghum bicolor* (L.) Moench), wheat (*Triticum aestivum* (L.)) and barley (*Hordeum vulgare* (L.)) [16]. It accounts for 5% of the total area allotted to cereal production in Ethiopia [17]. Finger millet is grown in more than 1.8 million households on more than 455,000 hectares of land in the northern, north-western, western, the Central Great Rift Valley and West Hararghe zones of Ethiopia [16]. In 2017 the total grain production was 1,017,059 tons, increasing by 87% in the preceding 20 years [17].

Despite the importance of finger millet for food security and livelihoods, its productivity is relatively low (2.47 t/ha) [16] in Ethiopia compared with the potential yield of the crop (6 t/ha) achieved under experimental conditions [18]. The low productivity of the crop in the country is attributable to a range of biotic and abiotic stresses and socio-economic constraints prevalent in the smallholder production systems in Ethiopia. Finger millet blast caused by *Magnaporthe grisea* (Barr) (teleomorph) is the most damaging disease, causing yield losses in the range of 7.32–54.07%, depending on climatic conditions and cultivar susceptibility [19]. Notable insect pests of the crop include grasshoppers (Caelifera) and shoot fly (*Atherigona soccata* (Rondani)) [15], pink stem borer (*Sesamia inferens* (Walker)), finger millet root aphid (*Tetraneura nigriabdominalis* (Sasaki)) and aphids (aphidoidea) [20,21]. Yield losses have been reported due to several insect pests such as termites (isopteran) (with a loss of 23%) [22], aphids (35.1%) [20] and pink stem borer (56%) [21]. Weeds cause



severe yield loss during the early growth stages of finger millet. In Ethiopia, yield losses of up to 73.5% have been reported due to weeds [23]. The most problematic weed species of finger millet in Ethiopia include *Digitaria ternata* (A. Rich.) Stapf, *Guizotia scabra* (Vis.) Chiov, *Cyperus rotundus* L. [23] and *Striga hermonthica* (Delile) Benth [24].

Recurrent drought stress associated with climate change is the leading constraint affecting finger millet production and other main crops in Ethiopia. The impact of drought stress on finger millet production depends on cultivar susceptibility, the onset date, the intensity and duration of the drought stress and the associated prevailing environmental conditions. Although finger millet is relatively drought-tolerant, 100% yield losses can be incurred due to intense and early onset of drought stress [25]. Supplementary irrigation, early planting and moisture conservation techniques such as mulching, zero tillage and tie ridging are often used to mitigate drought stress [26]. However, most smallholder farmers do not have access to irrigation and other resources to manage drought stress. Drought stress also significantly affects grain quality and yield components [27]. Hence, drought-tolerant varieties could be the most economical and environmentally friendly approach to controlling drought under smallholder production systems.

In Ethiopia, formal research on finger millet improvement started in the early 1980s [28]. In the last four decades, finger millet improvement activities in Ethiopia have focused on characterization and evaluation of locally collected and introduced germplasm for pure line selection. As a result, some 23 finger millet varieties have been registered and released for production [29]. Two varieties, Tadesse (KNE#1098) and Tessema (ACC#229469), were released with the beneficial traits of wide adaptability, high grain yield potential, good biomass and compact head shape. However, these varieties are late maturing, susceptible to insect pests and diseases, have relatively low human nutrition value and a seed shattering problem. The mean grain yield of improved finger millet varieties in Ethiopia is low at 2.7 t/ha [30], compared with 4.74 and 4.79 t/ha reported for Kenya [31] and India [32], respectively. Ethiopia is the primary centre of origin and diversity for finger millet [33]. The finger millet landraces grown by farmers are essential genetic resources that are known to hold useful genetic variation for desirable traits. Therefore, these landraces can be evaluated and selected for their desirable characteristics for new variety development, genetic analysis and gene discovery, leading to high yielding varieties that have all the essential farmer-preferred traits [34]. The finger millet production opportunities, farming systems, production constraints and preferred traits of the end-users are essential components for variety design and breeding strategies. Incorporating the needs and preferences of farmers would increase the adoption of new varieties of finger millet.

Farmers have a wealth of knowledge about their crops, farming systems and the constraints [35] that can be harnessed through a participatory rural appraisal (PRA). A PRA is a research tool used to gather useful information on farmers and their production systems for designing intervention strategies [36]. The PRA approach provides a platform for farmers and breeders to engage in information sharing actively. Plant breeders must understand farmers' situations and choices to design appropriate varieties to meet their needs. Several studies have used the PRA approach to gain insight into farmer production systems and varietal choices to prioritize breeding objectives, including in tef [37], sorghum [38], wheat [39]), pearl millet (*Pennisetum glaucum* (L.) R. Br) [40] and finger millet [41]. For example, drought is the major production constraint of finger millet in Eastern Uganda, according to Owere et al. [24], and in sorghum production in Ethiopia [42,43]. Similarly, a lack of access to improved seeds of groundnut [44] and sesame [34], a lack of improved varieties of sorghum [43] and a shortage of arable land and poor soil fertility in sorghum [42] were also identified as production limiting factors in Ethiopia. Likewise, a lack of improved finger millet and sorghum varieties in Uganda [24,38] and limited access to fertilizers in pearl millet production in Burkina Faso [40] have also been documented as production constraints. However, no recent study has documented farmers' perceptions of production constraints and trait preferences in finger millet in Ethiopia. Therefore, the objective of this study was to document finger millet production opportunities, constraints

and farmer-preferred traits in Ethiopia to set breeding goals and guide variety design in a finger millet improvement program.

## 2. Materials and Methods

### 2.1. Description of the Study Areas

The study was conducted in 2021 in the following two finger millet growing regions in Ethiopia: the Southern Nation Nationalities People Region (SNNPR) and Oromia (Figure 1). In the SNNPR, two districts, namely, Atote Ulo and Wera, were selected, while four districts (Shala, Siraro, Habro and Daro Lebu) were identified in the Oromia region for the study (Figure 1; Table 1). The geographical and climatic information for the study areas is presented in Table 1 [45]. The study areas fell within the mid to high altitude range between 1200 and 2400 m above sea level. The temperatures (°C) ranged between 12.5 and 29.1 °C with moderate to high mean annual rainfall of between 781.8 and 1103.6 mm year$^{-1}$.

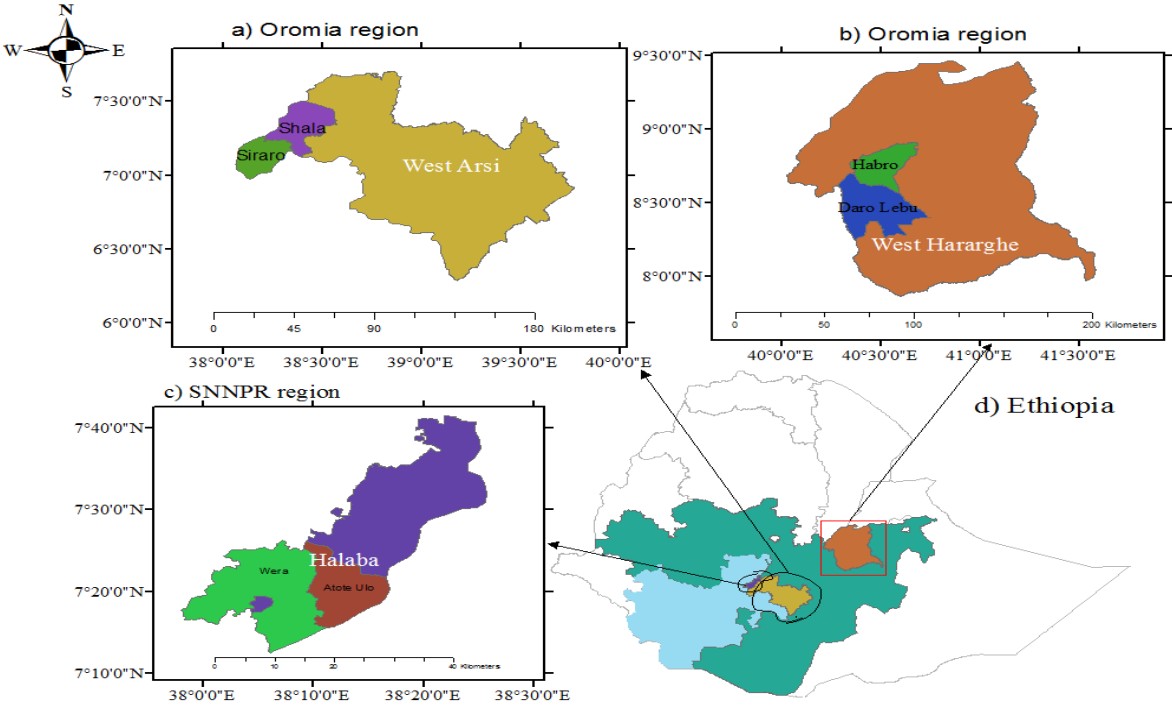

**Figure 1.** Geographical location of the study areas showing the regions, districts and zones.

**Table 1.** Descriptions of the study areas and number of sampled farmers for interviews and focused group discussion.

| Regions | Zones | Districts | Peasant Association | Altitudinal Ranges (m.a.s.l.) | Daily Mean Temperature Ranges (°C) | Annual Rainfall (mm year$^{-1}$) | No. of Interviewees | | | No. of Focused Group Discussants | | |
|---|---|---|---|---|---|---|---|---|---|---|---|---|
| | | | | | | | Male | Female | Total | Male | Female | Total |
| Oromia | West Arsi | Shala | Awara Gema | 1500–1900 | 12.9–26.7 | 781.8 | 10 | 10 | 20 | 08 | 07 | 15 |
| | | | Fendi Ejersa | | | | 15 | 05 | 20 | 11 | 04 | 15 |
| | | Siraro | Boye Awarakasa | 1500–2075 | 12.5–27.2 | 783.0 | 19 | 01 | 20 | 14 | 01 | 15 |
| | | | Damini Leman | | | | 18 | 02 | 20 | 13 | 02 | 15 |
| | West Hararghe | Habro | Gadisa | 1500–2400 | 13.4–28.3 | 1103.6 | 19 | 01 | 20 | 14 | 01 | 15 |
| | | | Kufa Kas | | | | 16 | 04 | 20 | 12 | 03 | 15 |
| | | Daro Lebu | Gelma Jeju | 1200–2000 | 14.1–29.1 | 1076.8 | 16 | 04 | 20 | 12 | 03 | 15 |
| | | | Oda Leku | | | | 20 | 01 | 20 | 14 | 01 | 15 |
| SNNPR | Halaba | Atote Ulo | 1st Ansha | 1800–1950 | 13.1–27.2 | 787.3 | 16 | 04 | 20 | 10 | 05 | 15 |
| | | | Girura Bucho | | | | 15 | 05 | 20 | 11 | 04 | 15 |
| | | Wera | Gedeba | 1700–2300 | 13.7–27.3 | 840.8 | 17 | 03 | 20 | 12 | 03 | 15 |
| | | | Kufe | | | | 19 | 01 | 20 | 13 | 02 | 15 |
| | | | Total | | | | 199 | 41 | 240 | 144 | 36 | 180 |

Note: m.a.s.l. = meters above sea level; SNNPR = Southern Nation Nationality and Peoples Region.

### 2.2. Sampling Procedures

A multi-stage sampling procedure was used to ensure a good representation of finger millet growers and the diverse agro-ecological zones in Ethiopia. Hence, a purposive sampling procedure was used to capture representative finger millet producing areas, major production opportunities and constraints and different socio-economic challenges. For the study two finger millet growing regions, SNNPR and Oromia, were selected. In the SNNPR region, the study was conducted in the Halaba Zone from which two districts were selected, namely, Atote Ulo and Wera. From each district, two peasant associations (PAs) were selected. A peasant association is locally referred to as 'Kebele', which is the smallest unit of administration in Ethiopia. This sampling provided a total of four PAs, including 1st Ansha and Girura Bucho (from Atote Ulo district) and Gedeba and Kufe (Wera district). The study was conducted in two zones in the Oromia region, including West Arsi Zone and West Hararghe Zone. From each zone two districts were selected (Siraro and Shala from West Arsi Zone and Habro and Daro Lebu from West Hararghe Zone). From Shala district, two PAs were selected (Awara Gema and Fendi Ejersa), two PAs from Siraro district (Boye Awarakasa and Damini Leman), two PAs from Habro district (Gadisa and Kufa Kas) and two PAs from Daro Lebu district (Gelma Jeju and Oda Leku). Participant farmers were randomly selected to represent the various wealth, gender and age group in the finger millet production community. Therefore, in each PA, 20 and 15 farmers were selected for face-to-face interview and focused group discussion (FGD), respectively, making a total of 240 and 180 participant farmers (Table 1).

### 2.3. Data Collection

Before data collection, enumerators were trained to ensure effective and efficient interviews and focus group discussions. The questionnaire was prepared in English and administered after translation to the local languages through trained enumerators. In the Oromia region, the local languages, namely, Afaan Oromo and Amharic, were used interchangeably to improve communication among researchers, enumerators and respondents, whereas in SNNPR, only the Amharic language was used. The questionnaires were pretested on a few respondents to improve clarity. Two breeders, a socio-economist, a pathologist and an agronomist were involved in facilitating and collecting data. Both primary and secondary data were collected for this study. Semi-structured questionnaire and FGD were used to collect the farmers' responses based on their 2020 finger millet farming experience. Semi-structured questionnaires were used to collect the following data: demographic attributes of the households, crops produced, roles of finger millet, improved varieties and local landraces, cropping system, seed systems, production constraints, drought coping mechanisms and farmers' varietal and trait preferences. FGDs were held to complement and confirm data gathered through interviews. The discussion topics for FGD were crops produced in the study areas, various roles of finger millet, improved varieties and local landraces, cropping system, seed systems, crop production constraints, coping mechanisms for drought, farmers' varietal and trait preferences and cost and cash income from finger millet production. Secondary data such as long-term weather data were collected from the National Metrological Agency of Ethiopia, while altitude, major crops grown and their area coverage and productivity were collected from the respective district Bureau of Agriculture.

### 2.4. Data Analysis

The qualitative data collected were coded into a suitable category and captured with quantitative data across the variables. Both data sets were subjected to data analysis using the Statistical Package for Social Sciences (SPSS) version 23 [46]. Descriptive statistics such as frequencies and percentages were computed using the cross-tabulation procedure. Significant tests were done with the chi-square test for qualitative and quantitative data sets. Contingency chi-square tests were employed to make statistical inference at the 0.05 level of significance to assess the relationship among variables. Conversely, the quantitative

data for cost-benefit analysis were summarized using Microsoft excel to calculate the ratios. Regarding the finger millet production constraints and farmers' trait preferences, they were labelled and tallied in a matrix, both in rows and columns, and the scores were obtained from pair-wise ranking based on one-to-one comparisons. The scores are equivalent to the frequency of respondents. Lastly, the scores were counted and used to conduct chi-square analysis and principal component analysis (PCA) for finger millet production constraints and farmers' trait preferences in the same order. PCA plots were developed to summarize the interrelationships of farmers' trait preference and their order of importance. Plots were done using the "FactorMineR" procedure [47] of R studio [48].

### 2.5. Cost Benefit Ratio Analysis

To appraise the monetary values of finger millet and other major crops production, the benefit to cost ratios were computed based on data collected in the study districts. The benefit to cost ratios were computed following the procedure of Adhikari [49] and Abraha et al. [37]. Microsoft excel was used to summarize the quantitative data sets of the different variables for the cost-benefit analysis.

$$Cost\ benefit\ ratio = \frac{\textbf{Total income}}{\textbf{Total production cost}}$$

Note: the total income included grain and straw sale, while the total production costs included the costs of seeds for planting, fertilizers, labour for land preparation, weeding, hoeing, thinning, harvesting and threshing.

### 3. Results

#### 3.1. Demographic Attributes

The demographic attributes of respondents and their households were summarized during the interviews (Table 2). There were significant differences ($X^2 = 17.8$; $p < 0.05$) in gender representation among the different districts. The majority (82.9%) were male farmers across all the study districts. The highest (15) and lowest (3) frequencies of female farmers were interviewed at Shala and Siraro, respectively.

**Table 2.** Proportion of respondents' gender, age, family size and level of education in the study districts.

| Variables | Categories | Districts | | | | | | Frequency | Percent |
| --- | --- | --- | --- | --- | --- | --- | --- | --- | --- |
| | | Atote Ulo | Wera | Shala | Siraro | Habro | Daro Lebu | | |
| Gender of household head | Female | 9 | 4 | 15 | 3 | 5 | 5 | 41 | 17.1 |
| | Male | 31 | 36 | 25 | 37 | 35 | 35 | 199 | 82.9 |
| | Chi-square test | | $X^2 = 17.8$ | | | df = 5 | | *p*-value = 0.003 | |
| Age of household head (year) | 18–40 | 29 | 31 | 24 | 31 | 33 | 26 | 174 | 72.5 |
| | 41–50 | 8 | 7 | 14 | 8 | 7 | 10 | 54 | 22.5 |
| | >50 | 3 | 2 | 2 | 1 | 0 | 4 | 12 | 5 |
| | Chi-square test | | $X^2 = 11.0$ | | | df = 10 | | *p*-value = 0.358 | |
| Number of children | ≤2 | 5 | 18 | 6 | 5 | 13 | 5 | 52 | 21.7 |
| | 3–5 | 14 | 8 | 7 | 11 | 19 | 13 | 72 | 30 |
| | ≥6 | 21 | 14 | 27 | 24 | 8 | 22 | 116 | 48.3 |
| | Chi-square test | | $X^2 = 38.4$ | | | df = 10 | | *p*-value = 0.000 | |
| Educational status of household head | Illiterate | 12 | 5 | 16 | 4 | 16 | 6 | 59 | 24.6 |
| | Read and write | 0 | 2 | 5 | 2 | 4 | 3 | 16 | 6.7 |
| | Grade 1–5 | 16 | 10 | 7 | 20 | 7 | 15 | 75 | 31.3 |
| | Grade 6–8 | 4 | 5 | 6 | 11 | 6 | 12 | 44 | 18.3 |
| | High school | 5 | 10 | 6 | 2 | 6 | 3 | 32 | 13.3 |
| | College | 3 | 8 | 0 | 1 | 1 | 1 | 14 | 5.8 |
| | Chi-square test | | $X^2 = 66.1$ | | | df = 25 | | *p*-value = 0.000 | |

Note: $X^2$ = chi-square test, df = degree of freedom, *p*-value = probability value.

The age groups of farmers did not show significant differences across the sampled districts ($X^2$ = 11.0, *p*-value = 0.36), with the majority of respondent farmers (72.5%) being between 18 and 40 years old. Only 5% of the farmers were older than 50 years, with none of the famers at Habro older than 50 years. There were significant differences ($X^2$ = 38.4; *p*-value = 0.000) in family sizes of respondents across the districts. Almost half (48.3%) of the respondents had households with more than five children. Habro and Wera districts had the lowest frequencies of farmers with more than five children, while Atote Ulo, Daro Lebu and Siraro had the lowest number of farmers with less than two children.

There were significant differences across districts ($X^2$ = 66.1, *p*-value = 0.000) in the levels of education. The highest proportion of farmers (31.3%) had attended school grades 1–5, while 24.6% had not attended any level of formal education. The highest frequency of respondents both with high school and college education was found at Wera at 10 and eight, respectively (Table 2).

### 3.2. Major Crops Grown in the Study Areas

Maize, tef and finger millet were the most important cereal crops grown in the study districts except in Daro Lebu and Habro, where sorghum was the most important and widely grown crop. There were significant differences among districts for a total area of production of finger millet ($X^2$ = 20.3, *p*-value = 0.03), maize ($X^2$ = 96.8, *p*-value = 0.000) and tef ($X^2$ = 28.5, *p*-value = 0.002). Similarly, significant variations were observed among districts for productivity of finger millet ($X^2$ = 64.392, *p*-value = 0.00), maize ($X^2$ = 34.255, *p*-value = 0.000), tef ($X^2$ = 31.862, *p*-value = 0.000) and sorghum ($X^2$ = 23.424, *p*-value = 0.009). The majority of the respondents allocated <0.25 ha of agricultural land each to finger millet (68% respondents), tef (51%) and sorghum (67%) production. About 10% of the respondents in Shala, Atote Ulo and Habro allocated a sizeable amount of land (>0.5 ha) to finger millet production. Conversely, about 83% of respondents in Siraro had a smaller land allocation (<0.25 ha) for finger millet (Table 3; Figure 2). Unlike the other main crops grown in the districts, nearly half of the respondents (46%) allocated large areas (>0.5 ha) of farmland to maize. Only 31% of the farmers in the study areas allocated <0.25 ha for maize production.

**Table 3.** Proportion (%) of respondents' farmland size (ha) allocation and productivity of major crops in the study districts during 2020/21 cropping season.

| Districts | Crops | | | | | | | | | | | |
|---|---|---|---|---|---|---|---|---|---|---|---|---|
| | Finger Millet | | | Maize | | | Tef | | | Sorghum | | |
| | Production Area (ha) of Crops and Proportion of Respondents (%) | | | | | | | | | | | |
| | <0.25 ha | 0.25–0.5 ha | >0.5 ha | <0.25 ha | 0.25–0.5 ha | >0.5 ha | <0.25 ha | 0.25–0.5 ha | >0.5 ha | <0.25 ha | 0.25–0.5 ha | >0.5 ha |
| Shala | 53 | 38 | 10 | 15 | 30 | 55 | 46 | 43 | 11 | 100 | — | — |
| Siraro | 63 | 28 | 10 | 8 | 36 | 56 | 48 | 29 | 23 | 100 | — | — |
| Atote Ulo | 70 | 20 | 10 | 16 | 21 | 63 | 24 | 38 | 38 | — | — | — |
| Wera | 83 | 18 | — | 18 | 30 | 53 | 60 | 28 | 13 | 50 | 50 | — |
| Habro | 80 | 18 | 3 | 89 | 5 | 5 | 75 | 15 | 10 | 65 | 26 | 10 |
| Daro Lebu | 63 | 38 | — | 92 | 4 | 4 | 85 | 8 | 8 | 62 | 29 | 10 |
| Mean (%) | 68 | 26 | 5 | 31 | 24 | 46 | 51 | 30 | 19 | 67 | 25 | 8 |
| Chi-square | $X^2$ = 20.3, df = 10, *p*-value = 0.03 | | | $X^2$ = 96.8, df = 10, *p*-value = 0.000 | | | $X^2$ = 28.5, df = 10, *p*-value = 0.002 | | | $X^2$ = 4.6, df = 10, *p*-value = 0.800 | | |

**Table 3.** *Cont.*

| Districts | Finger Millet | | | Maize | | | Tef | | | Sorghum | | |
|---|---|---|---|---|---|---|---|---|---|---|---|---|
| | Production Area (ha) of Crops and Proportion of Respondents (%) | | | | | | | | | | | |
| | <0.25 ha | 0.25–0.5 ha | >0.5 ha | <0.25 ha | 0.25–0.5 ha | >0.5 ha | <0.25 ha | 0.25–0.5 ha | >0.5 ha | <0.25 ha | 0.25–0.5 ha | >0.5 ha |
| | Productivity (t/ha) of crops and proportion of respondents (%) | | | | | | | | | | | |
| Districts | <1.5 t/ha | 1.5–3.0 t/ha | >3 t/ha | <1.5 t/ha | 1.5–3.0 t/ha | >3 t/ha | <1.5 t/ha | 1.5–3.0 t/ha | >3 t/ha | <1.5 t/ha | 1.5–3.0 t/ha | >3 t/ha |
| Shala | 25 | 50 | 25 | 21 | 29 | 50 | 100 | – | – | 75 | 25 | – |
| Siraro | 58 | 33 | 10 | 21 | 64 | 15 | 100 | – | – | 100 | – | – |
| Atote Ulo | 8 | 58 | 35 | 3 | 40 | 58 | 87 | 14 | – | 57 | 14 | 29 |
| Wera | 3 | 33 | 64 | 5 | 47 | 47 | 97 | 3 | – | 14 | 43 | 43 |
| Habro | 11 | 61 | 29 | – | 30 | 70 | 63 | 38 | – | 15 | 44 | 41 |
| Daro Lebu | 38 | 38 | 24 | 13 | 33 | 53 | 85 | 15 | – | 18 | 55 | 27 |
| Mean (%) | 24 | 45 | 31 | 11 | 42 | 48 | 90 | 10 | – | 28 | 41 | 31 |
| Chi-square | $X^2$ = 64.392, df = 10, *p*-value = 0.000 | | | $X^2$ = 34.255, df = 10, *p*-value = 0.000 | | | $X^2$ = 31.862, df = 5, *p*-value = 0.000 | | | $X^2$ = 23.424, df = 10, *p*-value = 0.009 | | |

Note: $X^2$ = chi-square, df = degree of freedom, t/ha = ton per hectare and *p*-value = probability value.

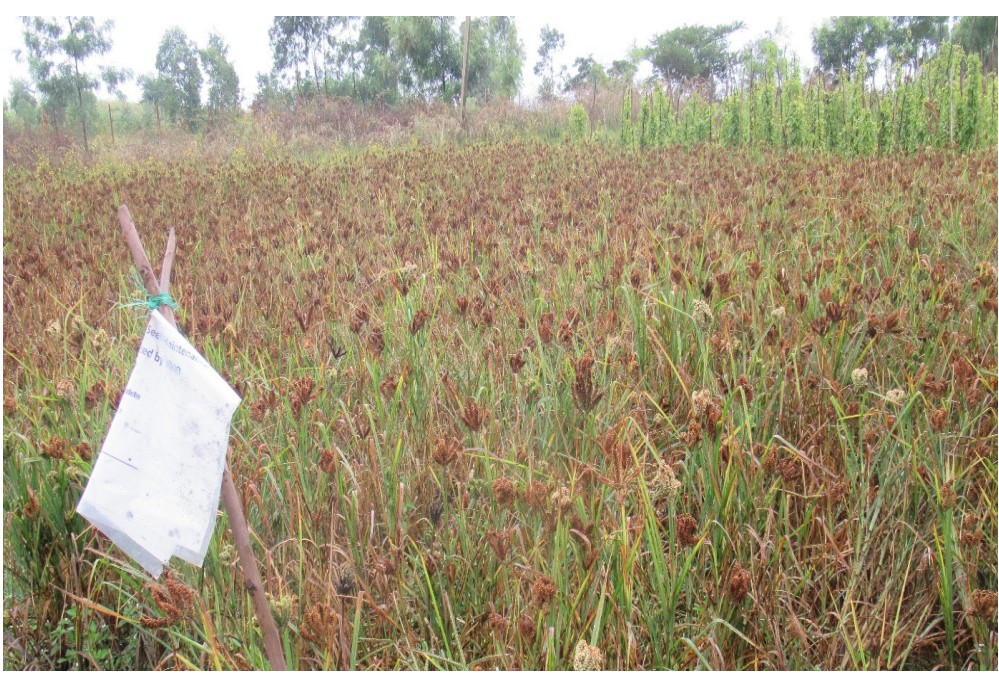

**Figure 2.** Finger millet seed production in western Ethiopia (photo H. Shimelis).

The majority of the respondent farmers reported yields in the range of 1.5–3.0 t/ha for finger millet and sorghum. A higher proportion of respondent farmers (90%) reported yields <1.5 t/ha for tef. Some 48% respondents reported yields >3.0 t/ha for maize. Farmers in Wera (64%) and Atote Ulo (35%) districts achieved finger millet yields of >3 t/ha (Table 3) due to the favorable growing conditions. The use of different crop management methods such as weed management practices, crop rotation and row planting are the most favorable growing conditions and essential drivers for high yield gains. Furthermore, farmers in these districts had access to improved seed through the Bureau of Agriculture and research institutions, which allowed higher yield gains.

### 3.3. Roles of Finger Millet Production in the Study Areas

Finger millet is a multi-purpose crop in the study areas. It is used for household consumption, cash income, feed, construction material and combinations of the various roles of the crop. The roles of finger millet significantly varied across the study districts ($X^2$ = 101.55; *p*-value = 0.00) (Table 4). A relatively higher number of respondent farmers (38.8%) used finger millet for a combination of food, feed and cash, followed by food and feed (14.5%), food and cash (13.7%), food (11.5%) and food, cash, feed and construction material (9.7%). About 11.5% of all the respondent farmers reported using finger millet for food only, while 19.4% in Shala district reported multiple purpose uses. A relatively higher number of farmers (22.5%) in Siraro district used the crop for food only, followed by food and feed (20%). Higher proportions of respondent farmers in Wera (65%), Atote Ulo (55%), Habro (40%) and Daro Lebu (38.9%) used finger millet for a combination of food, feed and cash (Table 4).

**Table 4.** Proportion (%) of farmers who grow finger millet for various roles in the study areas.

| Roles of Finger Millet | Districts | | | | | | Mean (%) | $X^2$ | df | *p*-Value |
|---|---|---|---|---|---|---|---|---|---|---|
| | Shala | Siraro | Atote Ulo | Wera | Habro | Daro Iebu | | | | |
| Food | 8.3 | 22.5 | 2.5 | 5.0 | 14.3 | 16.7 | 11.5 | | | |
| Feed | — | — | — | — | — | — | — | | | |
| Cash | — | — | — | — | — | — | — | | | |
| Food and feed | 13.9 | 20.0 | 15.0 | 2.5 | 14.3 | 22.2 | 14.5 | | | |
| Food and cash | 8.3 | 10.0 | 5.0 | 7.5 | 31.4 | 22.2 | 13.7 | 101.55 | 35 | 0.00 |
| Food and construction material | 5.6 | 2.5 | — | — | — | —- | 1.3 | | | |
| Food, feed and cash | 13.9 | 17.5 | 55.0 | 65.0 | 40.0 | 38.9 | 38.8 | | | |
| Food, feed and construction material | 11.1 | 12.5 | 5.0 | 5.0 | — | — | 5.7 | | | |
| Food, cash and construction material | 19.4 | 2.5 | 7.5 | — | — | — | 4.8 | | | |
| Food, income, feed and construction material | 19.4 | 12.5 | 10.0 | 15.0 | — | — | 9.7 | | | |

Notes: $X^2$ = chi square test; *p*-value = probability value, df = degree of freedom.

### 3.4. Socio-Economic and Environmental Factors Affecting Finger Millet Production in the Study Areas

Table 5 outlines the different constraints affecting finger millet production as perceived by the farmers. Constraints to finger millet production showed significant differences across the study districts ($X^2$ = 100.5; *p*-value = 0.00) (Table 5). About 41.5% of farmers reported drought stress as the foremost constraint affecting finger millet production, followed by a lack of improved varieties (12.9%), a lack of financial resources to purchase inputs (11.3%), land size limitations (10.8%), and limited access to seed (10.0%), shortage of fertilizers (5.4%), poor soil fertility (4.6%), shortage of draught power (1.3%) and labour shortage (1.3%).

**Table 5.** Proportion of farmers (%) who ranked the constraints to finger millet production in six districts of Ethiopia.

| Constraints | Districts | | | | | | Mean (%) | df | X² | *p*-Value |
|---|---|---|---|---|---|---|---|---|---|---|
| | Atote Ulo | Daro Lebu | Habro | Shala | Siraro | Wera | | | | |
| Drought stress | 47.5 | 35.0 | 35.0 | 40.0 | 35.0 | 55.0 | 41.3 | | | |
| Lack of improved varieties | 7.5 | 30.0 | 12.5 | 10.0 | 7.5 | 10.0 | 12.9 | | | |
| Lack of financial resources | 15.0 | – | – | 30.0 | 22.5 | – | 11.3 | | | |
| Land size limitation | 10.0 | 5.0 | 25.0 | 2.5 | 12.5 | 10.0 | 10.8 | | | |
| Limited access to seed | 10.0 | – | 10.0 | 15.0 | 12.5 | 12.5 | 10.0 | | | |
| Shortage of fertilizers | 2.5 | 15.0 | 2.5 | – | 10.0 | 2.5 | 5.4 | 45 | 100.5 | 0.000 |
| Poor soil fertility | – | 12.5 | 10.0 | 2.5 | – | 2.5 | 4.6 | | | |
| Shortage of draught power | – | 2.5 | 2.5 | – | – | 2.5 | 1.3 | | | |
| Labour shortage | 2.5 | – | 2.5 | – | – | 2.5 | 1.3 | | | |
| High labour costs | 5.0 | – | 0.0 | – | – | 2.5 | 1.3 | | | |
| Mean (%) | 100 | 100 | 100 | 100 | 100 | 100 | 100 | | | |

df = degree of freedom; X² = chi-square; *p*-value = probability level.

### 3.5. Farmers' Trait Preferences of a Finger Millet Variety

Farmers' trait preferences of finger millet were assessed and compared using PCA (Table 6). The first three principal components with eigenvalues greater than 1.00 explained 85.3% of the total variability in the desirable attributes of finger millet. The first principal component (PC1) accounted for 44.0% of the total variation, while the second and third PCs explained 26.1 and 15.2% of the variation, respectively. High grain yield (0.99) had the highest positive loading value on PC1, followed by compact head shape (0.94), large head size (0.93), pleasing aroma and taste of food products (0.75) and 'enjera'-making quality (0.60). Tolerance to lodging (−0.89), tolerance to shattering (−0.88), high tillering ability (−0.70), brew-making quality (−0.68), medium plant height (−0.54) and disease resistance (−0.53) had negative contributions to PC1. Insect pest resistance (0.77), early maturity (0.72), large grain size (0.7) and drought and heat tolerance (0.61) accounted for the highest variation on PC2. The ease of harvest and threshing (−0.84) had a negative loading on PC2. Only tolerance to weeds (0.84) and high marketability (−0.91) made large contributions on the third PC.

**Table 6.** Principal components and their contributions to finger millet agronomic and quality attributes reported in six districts in Ethiopia.

| Variables | PC1 | Contribution | PC2 | Contribution | PC3 | Contribution |
|---|---|---|---|---|---|---|
| High grain yield | **0.99** | **12.50** | −0.09 | 0.16 | 0.02 | 0.01 |
| Large head size | **0.93** | **10.86** | −0.13 | 0.34 | 0.09 | 0.30 |
| Weed tolerance | 0.09 | 0.10 | 0.36 | 2.79 | **0.84** | **25.65** |
| Disease resistance | **−0.53** | **3.54** | −0.31 | 2.05 | 0.40 | 5.82 |
| Ease of harvest and threshing | −0.27 | 0.94 | **−0.84** | **14.87** | 0.36 | 4.85 |
| Large grain size | 0.64 | 5.10 | **0.70** | **10.29** | 0.33 | 4.02 |
| Compact head shape | **0.94** | **11.03** | 0.17 | 0.58 | 0.28 | 2.89 |
| Insect pest resistance | −0.57 | 4.04 | **0.77** | **12.65** | 0.27 | 2.69 |
| Tolerance to lodging | **−0.89** | **9.92** | −0.30 | 1.94 | 0.23 | 1.89 |
| Brew-making quality | **−0.68** | **5.87** | −0.61 | 7.91 | 0.13 | 0.57 |
| High tillering ability | **−0.70** | **6.23** | 0.59 | 7.47 | −0.06 | 0.11 |
| Early maturity | 0.59 | 4.34 | **0.72** | **10.87** | −0.08 | 0.26 |
| Pleasing aroma andtaste of food products | **0.75** | **7.13** | −0.57 | 6.91 | −0.18 | 1.19 |
| Drought and heat tolerant | −0.20 | 0.51 | **0.61** | **8.03** | −0.23 | 1.95 |
| Medium plant height | **−0.54** | **3.72** | 0.45 | 4.39 | −0.26 | 2.47 |

**Table 6.** *Cont.*

| Variables | PC1 | Contribution | PC2 | Contribution | PC3 | Contribution |
|---|---|---|---|---|---|---|
| **Tolerance to shattering** | **−0.88** | **9.65** | −0.09 | 0.18 | −0.36 | 4.74 |
| **'Enjera'-making quality** | **0.60** | **4.46** | −0.55 | 6.51 | −0.54 | 10.53 |
| **High marketability** | −0.07 | 0.06 | 0.31 | 2.08 | **−0.91** | 30.05 |
| **Eigenvalues** | 7.9 | | 4.7 | | 2.7 | |
| % of total variance | 44.0 | | 26.1 | | 15.2 | |
| **Cumulative variance (%)** | 44.0 | | 70.1 | | 85.3 | |

PC = principal component, bold face values denote high score values.

Figure 3 presents the variables and study areas where the variables are connected with the biplot origin through the line vectors. The plot shows that high grain yield has the smallest angle with large head size followed by compact head shape, pleasing aroma and taste of food products, 'enjera'-making quality, large grain size and early maturity. Furthermore, the variables mentioned above have an angle less than 90° with high grain yield. On the other hand, the rest of the variables have an angle greater than 90° with high grain yield.

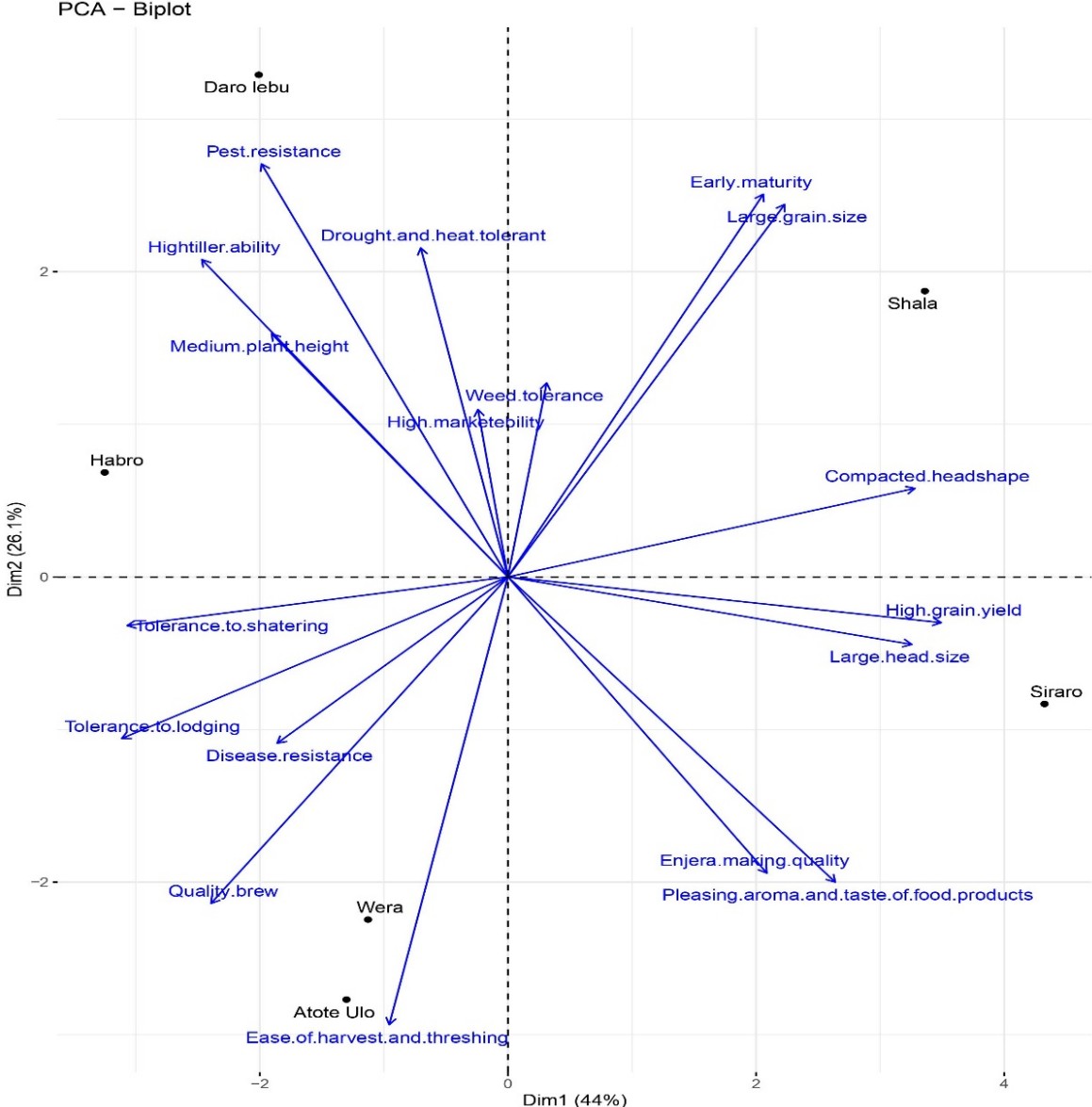

**Figure 3.** Biplot showing the interrelationships among the variables.

### 3.6. Crop Management Practices in Finger Millet Production

Respondent farmers reported the common crop management practices of finger millet. There were significant differences in finger millet growers' management practices across the districts (Table 7). About 40.3% of the respondents used a combination of hand weeding and chemical herbicides to control weeds, while 59.2% used hand weeding only. The largest proportion of respondent farmers (78.8%) used hand weeding in the Daro Lebu district. Shala and Siraro districts had the highest proportion of farmers at 65 and 55%, respectively, who controlled weeds using a combination of hand and chemical methods. Finger millet was planted as a sole crop by 97.0% of farmers. In all districts, a higher proportion of finger millet farmers (75.6%) practiced crop rotation with haricot bean, green pepper and potato. In Habro district, there were a lower proportion of farmers (20%) who practiced crop rotation. Direct field sowing was the major planting method of finger millet reported by 69.1% of respondents, followed by transplanting seedlings at 4–6 leaf stage. About 88.0% of the respondent farmers used row planting, while 12% practiced broadcasting. Some 51.4 and 15.4% of respondent farmers in Habro and Daro Lebu, respectively, used a broadcasting method of finger millet sowing (Table 7).

**Table 7.** The proportion (%) of respondents who used different crop management practices in finger millet production across the districts during 2020/20121 cropping season.

| Districts | Crop Management | | | | | | | | | | | |
| --- | --- | --- | --- | --- | --- | --- | --- | --- | --- | --- | --- | --- |
| | Weeding | | | Cropping System | | | Crop Rotation | | Transplanting | | Planting Methods | |
| | Hand Weeding | Chemical Herbicides | Hand Weeding and Chemical | Sole | Mixed | Sole and Mixed | Yes | No | Yes | No | Row | Broadcasting |
| Atote Ulo | 55.0 | – | 45.0 | 100.0 | – | – | 100.0 | 0.0 | – | 100.0 | 100.0 | – |
| Wera | 77.5 | – | 22.5 | 100.0 | – | – | 100.0 | 0.0 | – | 100.0 | 95.0 | 5.0 |
| Shala | 35.0 | – | 65.0 | 91.9 | 2.7 | 5.4 | 100.0 | 0.0 | 100.0 | – | 100.0 | – |
| Siraro | 45.0 | – | 55.0 | 97.5 | – | 2.5 | 72.5 | 27.5 | 80.0 | 20.0 | 97.5 | 2.5 |
| Habro | 67.5 | 2.5 | 30.0 | – | – | – | 20.0 | 80.0 | – | 100.0 | 48.6 | 51.4 |
| Daro Lebu | 78.8 | – | 21.2 | 100.0 | 5.4 | – | 58.8 | 41.2 | – | 82.5 | 84.6 | 15.4 |
| Mean (%) | 59.2 | 0.5 | 40.3 | 97.0 | 1.7 | 1.3 | 75.6 | 24.4 | 30.9 | 69.1 | 88.0 | 12.0 |
| Chi-square test | $X^2 = 30.7$ df = 10 $p$-value = 0.00 | | | $X^2 = 12.7$ df = 10 $p$-value = 0.24 | | | $X^2 = 111.3$ df = 5 $p$-value = 0.00 | | $X^2 = 203.0$ df = 5 $p$-value = 0.00 | | $X^2 = 70.4$ df = 5 $p$-value = 0.00 | |

Notes: $X^2$ = chi-square test; $p$-value = probability value, df = degree of freedom.

### 3.7. Finger Millet Varieties Grown and Sources of Seed

3.7.1. Attributes of Varieties Cultivated by the Farmers

There have been 20 finger millet varieties officially released in Ethiopia since 1998. However, only a few of these varieties are presently grown in the study areas (Table 8), such as Tadesse, Tessema, Axum, Meba and Bareda. However, late maturity, susceptibility to disease (head blast), insect pests and shattering problems were among the released varieties' major drawbacks (Table 8).

**Table 8.** Lists of released finger millet varieties and landraces, their preferred traits and drawbacks reported by respondent farmers in 2020/21 cropping season.

| Name of Variety or Designation | Preferred Traits | Drawbacks |
|---|---|---|
| Released Varieties | | |
| Tadesse (KNE#1098) | Easy to thresh, high yielding, medium plant height, lodging resistance, compact head, large grain size, high biomass, erect tillers, good for feed | Late maturing, susceptible to insect pests and diseases, shattering problem, low human health value |
| Tessema (ACC#229469) | High yield, compact head, high biomass, easy to thresh | Susceptible to insect pests and diseases, late maturing, low medicinal value |
| Axum | High yielding, drought tolerance | Susceptible to root rot disease, low human health value |
| Meba | Disease resistant | Low yielding |
| Bareda | High yield, good biomass | Low human health value |
| Landraces | | |
| Amaracha | Good food quality, insect and disease resistance, human health value | Low yield, susceptible to lodging, low biomass production |
| Dagusa | Good food quality, insect and disease resistant, human health | Difficult to thresh |
| Dima (red seed type) | Good food quality, insect and disease resistant, better in medicinal value | Low biomass |
| Dalecha (dark brown seed) | High tillering capacity, medicinal value | Low yielding |
| Ejeru | Lodging resistance, early and good 'enjera'-making quality | Susceptible to disease |
| Guracha (black seed) | Good for 'enjera' and high yielding | Susceptible to drought |
| Habesha | Good food quality | Low yield |
| White | High yield and good for 'enjera'-making | Susceptible to disease, late maturing |

Farmers also cultivated landrace or local finger millet varieties (Table 8). The main distinguishing features used in the selection of the local varieties were local names and seed colour. Respondent farmers preferred the landraces for their higher perceived nutritional and human health values than the released varieties. The farmers mentioned that the local finger millet varieties are also preferred for their tolerance to disease and insect pests. However, the local landraces are cultivated on small areas because they have low yield and biomass production, susceptibility to lodging and are difficult to thresh. The harvested seed from the local landraces is not true-to-type due to genetic admixtures.

3.7.2. Sources of Finger Millet Seed

There were significant differences ($X^2$ = 191.597, *p*-value = 0.000) among respondent farmer seed sources (Table 9). The Bureau of Agriculture (BOA) was the primary source of finger millet seed. About 49.1% of respondent farmers across the districts accessed seed from the BOA. In the Atote Ulo and Wera districts, 88 and 93% of respondents, respectively, used BOA as their seed source. The next important source of seed was farmer-to-farmer exchange. On average, 22% of the respondents used seed obtained from other farmers. Daro Lebu (reported by 50% of respondents) and Siraro (35%) had the highest frequencies of farmers who exchanged seed with other farmers. Research institutions such as the Ethiopian Institute of Agricultural Research (EIAR) and Oromia Agricultural Research Institute (OARI), local producers and self-saved seed were also mentioned as seed sources by 32, 28 and 22% of respondents at Shala, Siraro and Habro, in that order (Table 9).

**Table 9.** The proportion (%) of respondents and corresponding seed sources of finger millet varieties in the study districts in 2020/21 cropping season.

| Seed Sources | Districts | | | | | | Mean (%) |
|---|---|---|---|---|---|---|---|
| | Atote Ulo | Wera | Shala | Siraro | Habro | Daro Lebu | |
| Research institutions | 5.0 | — | 32.0 | — | 2.7 | — | 5.8 |
| Bureau of Agriculture | 87.5 | 92.5 | 35.50 | 22.5 | 35.1 | 15.8 | 49.1 |
| Local producers | 2.5 | 2.5 | 6.50 | 27.5 | 2.7 | 2.6 | 7.5 |
| Farmer-to-farmer seed exchange | 2.5 | — | 22.60 | 35.0 | 24.3 | 50.0 | 22.1 |
| Own saved seed | - | 5.0 | — | — | 21.6 | 18.4 | 7.5 |
| Cooperatives | 2.5 | — | — | — | — | — | 0.4 |
| Local market | — | — | — | 7.5 | 5.4 | 7.9 | 3.5 |
| Unknown source | — | — | 3.20 | 7.5 | 8.1 | 5.3 | 4.1 |
| Chi-square test | $X^2$ = 191.597, df = 35, *p*-value = 0.000 | | | | | | |

### 3.8. Cost Benefit Analysis of Major Crops Grown in the Study Areas

The economic importance of the major crops grown in the study areas was assessed through cost-benefit analysis. In this regard, the principal crops were compared concerning achieved yield (t/ha), total income realized from sales of grain and straw in United States dollars per hectare (USD/ha), total production costs (USD/ha), revenue (USD/ha) and benefit to cost ratios. The highest grain yield (3.00 t/ha) was obtained from finger millet followed by maize (2.93 t/ha) and sorghum (2.20 t/ha). Conversely, tef had the least yield (0.65 t/ha) followed by haricot bean (1.74 t/ha). The total income (USD/ha) generated from finger millet was the highest at 2139 USD/ha followed by sorghum (1612 USD/ha) and haricot bean (1033 USD/ha). Total income generated from haricot bean and maize sales were 1033 and 1003 USD/ha, respectively, lower than the average price of all crops (1329.8 USD/ha) (Table 10).

**Table 10.** Income, cost and cost-benefit analysis of finger millet and other major crop production in the 2020/21 cropping season in the study districts.

| Crops | Price of Grain (USD/ton) | Grain Yield (t/ha) | Income from Grain Sell (USD/ha) | Income from Straw Sell (USD/ha) | Total Income (USD/ha) | Total Production Cost (USD/ha) | Profit (USD/ha) | Benefit/Cost Ratio |
|---|---|---|---|---|---|---|---|---|
| Finger millet | 630.67 | 3.00 | 1892.00 | 247.00 | 2139.00 | 1249.00 | 890.00 | 1.71 |
| Haricot bean | 566.67 | 1.74 | 986.00 | 47.00 | 1033.00 | 268.00 | 765.00 | 3.85 |
| Maize | 303.75 | 2.93 | 890.00 | 113.00 | 1003.00 | 743.00 | 259.00 | 1.35 |
| Sorghum | 646.36 | 2.20 | 1422.00 | 190.00 | 1612.00 | 689.00 | 923.00 | 2.34 |
| Tef | 1189.23 | 0.65 | 773.00 | 89.00 | 862.00 | 509.00 | 353.00 | 1.69 |
| Mean | 667.30 | 2.10 | 1192.60 | 137.20 | 1329.80 | 691.60 | 638.00 | 2.19 |

The total cost of finger millet production (1249 USD/ha) was at least twice as high as the average cost of production of all other crops grown (691.6 USD/ha) in the study areas. Sorghum was the most profitable crop, with an average profit of 923 USD/ha, followed by finger millet (890 USD/ha). Tef growers realized significantly lower profits of 353 USD/ha, while maize growers attained the least profit of 259 USD/ha (Table 10).

The principal crops were also compared in terms of benefit/cost ratio. On average, haricot bean producers with a higher benefit to cost ratio of 3.85 had the highest benefit, followed by sorghum growers (2.34). The benefit/cost ratios for finger millet, tef and maize

were 1.71, 1.69 and 1.35, respectively. The significant costs of finger millet production were the costs of the seed for planting, fertilizers, labour for land preparation, weeding, hoeing, thinning, harvesting and threshing.

### 3.9. Cultural Practices to Cope with Low Moisture Stress

The respondent farmers developed a range of agronomic solutions to finger production challenges, mainly drought stress (Table 11). As a result, farmers in the study areas used various cultural methods to cope with moisture stress. Ploughing varied in terms of frequency, depth and date as a moisture stress coping strategy. Hoeing at the right stage, weed control and supplemental irrigation, if available, were also used to mitigate moisture stress. Moreover, adjustment of sowing dates, tie ridging and relatively deep sowing were used to manage moisture stress. Farmers planted at higher than recommended seeding rates to attain optimal plant populations. Varietal selection, application of inorganic fertilizers and the use of mulching and cattle dung were also used to manage moisture stress (Table 11).

**Table 11.** Various methods used by finger millet growers to cope with moisture stress, reported during the focus group discussion.

| Methods to Cope with Moisture Stress | Perceived Advantages |
| --- | --- |
| High ploughing frequency before the onset of rainfall | Assists in infiltrating the available soil moisture, exposure to sunlight of eggs of insect pests present in the soil. |
| Deep ploughing by using tractor | Improves moisture-holding capacity of the soil, exposure to sunlight of eggs of insect pests present in the soil |
| Early ploughing and land preparation as soon as the onset of the first rain shower after harvesting | Effective use of the available soil moisture |
| Hoeing at the right stage | Maintains the available soil moisture |
| Weed control | Protects the crop from the competition of the soil moisture and other nutrients |
| Irrigation if available | Provides supplemental moisture required by the crop |
| Adjustment of sowing date | Manages flowering time so as not to coincide with drought times and utilizes the available soil moisture |
| Sowing in tie ridging | Holds available soil moisture |
| Row planting | Manages the appropriate plant population |
| Sowing the seed relatively deep in the soil | Assists the seed to access the available soil moisture for germination |
| Use of higher seed rate than the recommended one | Assists to get the required plant population during low moisture period |
| Soil mulching using different grass species | Increases soil fertility and water holding capacity and lowers soil temperature |
| Use of cattle dung and application of urea fertilizer after the first weeding and when there is a relatively good soil moisture | Increases soil fertility and moisture-holding capacity and provides healthy and vigorous crop to cope with low moisture stress period |
| Varietal selection | Better and cheap alternative to alleviate the problem of low moisture stress |

## 4. Discussions

### 4.1. Demographic Attributes

The demographic characteristics of the respondents were documented (Table 2) because they influence farming practices, intervention strategies and technology adoption among farming communities. The most significant proportion of respondents were male, which is concomitant with the fact that most households in the study area were male headed (Table 2). Patriarchy is dominant in the study area, with a negligible number of females having decision-making powers. The disenfranchisement of females, as discovered in other PRA studies, also reflects their peripheral roles in decision-making in agricultural activities and their ongoing exclusion from social services such as training and agricultural extension services [50], despite their active participation in farming operations such as ploughing, weeding and harvesting.

In terms of age, most of the interviewed farmers were within the active age group of 18–40 years (Table 2). This group consists of young and middle-aged adults that participate in the economy by providing labour and engaging in economic activities, such as trade, and in decision-making. Mulalem and Melak [22] also identified this group as vital for agricultural functioning as an active, productive age group. The young adults in this group can adopt new agricultural technologies, given their high literacy level and lack of prior experience [51]. The middle-aged adults in the active productive group were involved in decision-making and influenced choices of agricultural technologies, which in turn have an impact on crop production and productivity [36].

The respondent farmers had large families of more than five children per household, which positively impacts the provision of labour for crop production but is a concern for food insecurity in the study districts. Large families provide readily available labour for farming activities in subsistence farming systems because the farmers cannot afford to hire external labour [43]. Smaller households struggle to implement essential activities such as ploughing and weeding, given that most operations in smallholder farming are manual. Provision of labour is also related to the age of family members. Families composed of mostly young children struggle to provide the required labour. However, large families require more significant amounts of food for sustenance, and the risk of food insecurity increases in subsistence farming where crop productivity is generally low. Tadele [52] noted that large families have an adverse impact on food security, especially in Africa, where the population growth rate is very high.

The low literacy levels among the sampled farmers are of concern, especially for the successful introduction of new technologies and dissemination of information. A low level of education has been identified as a significant factor leading to poor adoption of agricultural technologies and access to information in rural and smallholder farming communities. Interventions such as farmer training and provision of information have less impact on agriculture systems where farmers have low levels of literacy [50,53]. Farmers who have a higher level of literacy are likely to adopt improved technologies and improve their farming practices for higher crop productivity and have the potential to engage in more profitable markets or negotiate for better prices with service providers [50,54].

### 4.2. Dominant Crops Cultivated in the Study Areas

Crop production was dominated by maize and finger millet (Table 3), consistent with previous reports showing that smallholder farmers cultivated mainly maize and other cereals crops [55]. The land allocated for finger millet production by a household was equivalent to the national average of 0.25 ha [17], showing that the selected study sites could represent finger millet production systems in Ethiopia. The production of finger millet is essential for mitigating the impact of drought stress on food security. Finger millet is more drought tolerant than crops such as maize. However, the dominance of cereals is a concern for nutritional security. Cereal-based diets are carbohydrate-rich, leading to hidden hunger caused by deficiencies in essential nutrients such as specific amino acids, minerals and vitamins. Finger millet, and sorghum to an extent, are high in micronutrients,

and farmers in the study areas get the various health benefits in their food sources from the two crops. The farmers reported trading their grains for cash income generation to buy other foods containing proteins, vitamins and minerals to supplement their cereal diets. However, low productivity and a lack of surplus grain yield frequently limit potential income generation. Yields below 1.5 t/ha have been commonly recorded in the study areas, below the national average for all crops. The dominance of maize in production systems of the study areas has been enhanced by its potential due to its early sowing dates, where green maize is grown to avert food shortages in the lean season (when the previous season's grain harvest becomes depleted but the new crop is not available), and its relative ease of marketing compared to finger millet or other cereals.

### 4.3. Various Uses of Finger Millet

Foremost, finger millet is used as a food crop in the study areas (Table 4). It is commonly ground into flour for making leavened bread known locally as 'enjera'. However, finger millet has relatively poor 'enjera'-making qualities and the farmers usually blend the finger millet flour with maize flour. Alternatively, finger millet is coarsely ground to make porridge. However, porridge made from finger millet is not common in Habro and Daro Lebu districts, where the farmers mentioned that they do not use finger millet to make porridge. Cultural differences and access to information influence the uses of finger millet. Training and awareness campaigns on the potential uses of finger millet and bio-fortification of finger millet could improve its utilization and contribute to food security.

Finger millet straw is also vital for livestock feed (Table 4). The farmers have small land holdings, and their livestock are raised on communal grazing lands. After harvest, the livestock are allowed into the fields to graze on crop residues. Most of the farmers in the study areas harvested the stover to feed the livestock when there was scarcity. While this stover's nutritional value and palatability are relatively low relative to a green fodder crop [56], its impact on animal health is vital given the lack of alternative grazing in the dry season. Mululam and Melak [22] reported that 69% of farmers in North-Western Ethiopia used finger millet straw for animal feed, while 12% used the straw as a construction material. Studies in China showed that the replacement of other straws like corn straw with finger millet straw improved the growth of sheep and was recommended in fattened lamb production [57].

### 4.4. Socio-Economic and Environmental Factors Affecting Finger Millet Production

While the ranking of the importance of production constraints varied across the districts, erratic rainfall, a lack of improved varieties, a lack of financial resources to procure inputs, land shortages, a limited supply of seeds of improved varieties, a lack of access to fertilizers and declining soil fertility were the most common challenges affecting finger millet production (Table 5). Erratic rains were also identified as a major production constraint in Kenya [58], Myanmar [59] and Ethiopia [60,61]. A lack of financial resources has been previously identified as the single most crucial socio-economic challenge affecting crop production in most sub-Saharan African countries [62]. Limited access to agricultural inputs such as fertilizers, pesticides and improved seeds is related to limited financial resources and has been widely reported in Africa [63] and, particularly, Ethiopia [53,64,65].

Smallholder farmers face a multitude of production constraints that limit crop productivity. Biotic and abiotic constraints, such as pests and diseases and declining soil fertility, may be mitigated with breeding for varieties with the necessary resistance or tolerance level to support crop production in stress-prone environments. On the other hand, socio-economic constraints can be rectified by implementing necessary policy changes, training intervention and improving extension services. Both policy regulations to improve the socio-economic environment and breeding are still lagging, which significantly compromises crop production in general, particularly finger millet.

### 4.5. Farmers' Trait Preferences of a Finger Millet Variety

The most desirable traits of finger millet were compared and their order of importance were assessed via PCA. The biplot shows the interrelationship among the variables. The cosine of the angle between the vectors of two variables is almost equal to the correlation coefficient between them [66]. The angle between the two variables is an indication of how closely or distantly related the variables are. Therefore, the smaller the angle between them the stronger the relationship they have and vice versa. Comparison of the angles in Figure 3 and the principal component analysis of Table 6 showed a high correspondence between them. Identifying farmer-preferred traits is an essential step for variety design and development. High grain yield, 'enjera'-making quality, large head size and compacted head shape can be prioritized in variety development to meet the aspirations of the farmers (Table 6). The inclusion of farmer-preferred traits in variety development is essential to promote cultivar adoption but also to mitigate production constraints. Traits such as insect pest and disease resistance and drought and heat tolerance are vital for inclusion in new varieties, given that the farmers alluded to the impact of biotic and abiotic constraints on finger millet production (Table 6). While the ranking of farmer-preferred traits varied across the districts, the identified traits were consistent and could potentially be pyramided into a single variety. After identifying farmer-preferred traits in finger millet, the next step would be to understand the genetic basis of the traits and devise suitable strategies for their improvement in new cultivars. Traits such as high grain yield and drought tolerance are quantitative traits that are difficult to improve due to their polygenic nature and high environmental variance. They will require the collection of diverse genetic resources for evaluation and selection to develop suitable varieties. For traits such as blast disease resistance, additive gene action has been predominant for finger millet and showed that progress would be made through recurrent selection [67].

Similarly, traits like high 'enjera'-making quality are likely to be governed by a few major genes; the selection process and variety development may be relatively easier and faster than high yield and drought-tolerant variety development. Given that farmers desired multiple traits in a single finger millet variety, breeding an ideal variety will not be a straightforward process. This process of soliciting information from farmers can be conducted periodically and iteratively at all stages of variety design to incorporate new ideas and insights and respond to changes in environment and lifestyle. Owere et al. [24] also reported that high grain yield, compact head shape and early maturity were the most preferred attributes of finger millet. Likewise, high yield, drought tolerance, early maturity and big heads were key farmer-preferred traits reported by Ojulong et al. [68] and Tracyline et al. [69].

### 4.6. Cropping Patterns of Finger Millet and its Management Practices

Weed control was one of the major tasks carried out by farmers, and the use of manual labour to control weeds is both inefficient and time-consuming (Table 7). The combination of herbicides and manual labour is more efficient but was limited by the farmers' shortage of inputs and lack of financial resources. Finger millet was planted as a sole crop, which agreed with another report showing that finger millet is commonly grown as a sole crop [9]. Unlike maize, which is sometimes intercropped with legumes, there are very few cases where finger millet is intercropped with legumes. The most common practice is to rotate finger millet with other cereals or legumes, which farmers in the study areas practiced. During group discussions, the respondents pointed out that finger millet was planted as a sole crop but in rotation with haricot bean and hot pepper. In addition to crop rotation, farmers used double cropping systems involving tef and haricot bean. However, the double cropping system was not possible with finger millet because the currently cultivated finger millet varieties are too late-maturing to fit into a double cropping system. Developing and deploying early maturing varieties would facilitate its inclusion in a double cropping system for enhanced food production.

### 4.7. Seed Source of Finger Millet

Currently, there is a poorly developed seed system industry for finger millet in Ethiopia. A significant dependence on BOA and farmer-to-farmer seed exchange is linked to poor access to seeds of improved varieties (Table 9). Farmer saved seeds are not pure, have low germination rates and often carry seed-borne diseases [70], contributing to yield losses. While the BOA was a seed source for most farmers, it often has limited supplies and cannot reach all the farmers simultaneously for planting. It is imperative that as breeding programs commence, they can be developed in parallel with a commercial seed system to ensure efficient and effective distribution. There are also few registered finger millet varieties in Ethiopia despite the importance of finger millet as a crop. This is concordant with previous reports on the neglect of traditional cereals in breeding programs compared to crops such as maize and wheat. Of the 23 released varieties, only five were in production, which begs the question why the farmers poorly adopt them. A possible reason is lack of farmer involvement in previous breeding programs that focused on product development with little regard to farmer input. Recently, most programs have developed varieties that were high yielding but lacked other vital and complementary attributes desired by farmers, leading to their rejection by the market. In this regard, Jerop et al. [51] reported that the significant seed sources of finger millet in Kenya were self-saved seed and the government extension program, which corroborate the findings of this study. Tsehaye et al. [9] reported that most farmers in Northern Ethiopia also used self-saved seed or obtained seed from the local informal market.

### 4.8. Cost-Benefit Analysis of Major Crops Grown in the Study Areas

The high cost of production for finger millet was probably driven up by the high labour costs for weeding due to its susceptibility to weed competition during its early stages of growth. Manual weeding was practiced at a higher frequency for finger millet than other crops, requiring more man hours and increasing production costs. In addition to weeding, harvesting and threshing of finger millet are tedious and labour-intensive. In general, weeding, harvesting and winnowing were the significant labour demanding tasks in finger millet production. Even though finger millet is a highly profitable crop (890 USD/ha), the respondents expressed reluctance to produce it on a large scale, citing the high labour requirements as an impediment. Higher labour requirements for finger millet production than other crops have been identified as a major deterrent to its production, productivity and market potential [40]. In India, the average cost of production for finger millet was estimated to be 544.3 USD/ha, with average yield productivity of 1.44 t/ha and a net profit of 138.1 USD/ha [71]. The cost-benefit ratio calculated for finger millet was similar to 1.05 reported by Adhikari [49] and within the range of 1.05–2.15 that was reported by Kaushal and Choudhary [71]. There is a need to increase the benefit to cost ratio to motivate the farmers to adopt finger millet production. Improved resistance to weeds, increased thresh ability and early maturity would reduce labour costs associated with the respective agronomic practices and encourage farmers to adopt the crop. Therefore, there is a need for finger millet improvement to deliver high yielding and farmer-preferred varieties to enhance the economic benefits of the crop. Maize is one of the major crops in Ethiopia, including in the study areas. Nevertheless, farmers are not deriving profits from the production and marketing of this crop due to various reasons. The primary reason is that, in the country, the grain prices of maize are unpredictable due to the high market supply during the production season. This condition is the major constraint for maize farmers given that most of them have access to the local markets to sell maize [72]. In addition, there are no adequate postharvest infrastructures in the country, including transport, storage and processing.

### 4.9. Cultural Methods to Cope with Low Moisture Stress

Production of drought-tolerant crops such as finger millet has been promoted as a strategy for climate change mitigation [73]. Farmers in the study areas were aware of

climate change, its adverse effects and possible mechanisms to cope with its effects. As a result, they used various strategies to cope with low moisture stress to minimize crop loss and food insecurity. These included various soil moisture conservation and soil fertility enhancement technologies (Table 11). The frequency, depth and period of ploughing and the timing of crop management practices such as planting, weeding, and adjusting plant population were used to mitigate the impact of moisture stress, with various levels of success. Similarly, during the period of low moisture stress, most farmers in South and North Welo grew early maturing sorghum to escape drought stress [42]. The breeding of short duration finger millet varieties would also help the crop to escape drought stress. Mulching and the use of tie ridges were practiced because these practices are commonly used for moisture conservation. Early planting, use of organic inputs, adoption of new tillage practices and applying tied ridges have been previously reported among strategies used by smallholder farmers to mitigate the impact of low soil moisture [74].

## 5. Conclusions

Finger millet is one of the staple food crops in Ethiopia, but its productivity is constrained by a range of biotic and abiotic stresses and socio-economic factors. Drought stress was considered to be the most important constraint in all the districts, followed by a lack of improved varieties, limited access to seed and a lack of financial resources. Land size limitations, poor soil fertility and a lack of access to fertilizers were also ranked important constraints affecting finger millet production. The most critical farmer-preferred traits in finger millet were high grain yield, compact head shape, 'enjera'-making quality, high marketability and early maturity. Therefore, to enhance finger millet productivity, plant breeding aimed at solving the above-mentioned production constraints and incorporating the farmer-preferred traits needs to be undertaken in Ethiopia.

**Author Contributions:** A.G., data curation, formal analysis, methodology, software, visualization, original draft, writing—review and editing.; H.S., conceptualization, funding acquisition, investigation, methodology, project administration, resources, supervision, validation, visualization, writing—review and editing.; M.L., funding acquisition, investigation, resources, supervision, writing—review and editing.; I.M., conceptualization, methodology, validation, writing—review and editing.; D.A.O., funding acquisition, resources, investigation, supervision, writing—review and editing. H.O.; resources, funding acquisition, visualization, writing—review and editing. All authors have read and agreed to the published version of the manuscript.

**Funding:** The authors wish to thank the International Crops Research Institute for the Semi-Arid Tropics (ICRISAT) for offering the fellowship and financial support to the first author through 'Harnessing Opportunities for Productivity Enhancement (HOPE II) for Sorghum and Millets in Sub-Saharan Africa' project (grant number OPP1198373), funded by the Bill and Melinda Gates Foundation (BMGF). The authors also thank ICRISAT for the support of the publication of this research work through the support of the project "Safeguarding crop diversity for food security: Pre-breeding complemented with Innovative Finance" which is funded by the Templeton World Charity Foundation, Inc. (TWCF0400) and managed by the Global Crop Diversity Trust.

**Institutional Review Board Statement:** Not applicable.

**Informed Consent Statement:** Not applicable.

**Data Availability Statement:** Not applicable.

**Acknowledgments:** Ethiopian Institute of Agricultural Research and Melkassa Agricultural Research Center are acknowledged for supporting this study and providing study leave for the first author. The University of KwaZulu-Natal is also gratefully acknowledged for the PhD study placement of the first author. National Sorghum and Millets Research Coordination of Ethiopia, National Metrological Agency of Ethiopia and Bureau of Agriculture of Halaba, West Arsi and West Hararghe Zones in Ethiopia are gratefully acknowledged for making this study possible. Finally, we thank all the farmers of the study area for sharing their valuable time and knowledge and for making this study possible.

**Conflicts of Interest:** The authors declare no conflict of interest.

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
