# Peer review of "Finger Millet Production in Ethiopia: Opportunities, Problem Diagnosis, Key Challenges and Recommendations for Breeding"

_sustainability, doi:10.3390/su132313463_

Round 1

Reviewer 1 Report

KEY MESSAGE

The manuscript is well English-written and contains substantial scientific and agronomic value about understanding Millet production systems, especially in Africa, an ascending tropical agriculture frontier, to cope with the demand for food from the growing world population. Although the manuscript contains several positive scientific contributions, adjustments in the description of the statistical analysis are required, and further analysis/visualization of the results.

COMMENTS

In lines 51-67: please add a statement about the agricultural importance of Ethiopia in the Africa continent.   

In lines 201-204: please better explain how qualitative and quantitative tests were performed. What is t analysis? Do you mean the t-student test? What was the null and alternative hypothesis? What was explicitly tested? What is the significance level assumed?  This paragraph needs to be improved substantially with further information about the analysis for clarity.

Would you please add the significance level in all tables related to tests?

It does not make sense p-value = 0; this would mean complete certainty. For instance, in Table 2, in the chi-square test related to the number of children, the p-value should be 3.2 × 10-5, not zero. Please, review all tests performed in the manuscript, and fix p-value = 0 cases.

Please, replace (p=) with (p-value=).

Please, a table with the correlation between the quantitative descriptors collected in the manuscript, and report on results/discussion where best judged.

Please, provide a figure with a geographical heatmap of the cost-benefit ratio calculated across the geographical regions and discuss in the result section where appropriate.

Please, generate a principal component analysis plot displaying how the variables mentioned in Table 6 are closely or distantly related and discuss where best judged.

In lines 252-255: what are the favorable growing conditions? Would you please provide further information about the key drivers of the high yield?

In lines 368-372: please discuss agricultural management practices used by the growers to justify maize as the least profitable crop.

Author Response

Comment 1: In lines 51-67: please add a statement about the agricultural importance of Ethiopia in the Africa continent.   

Response 1: Revised as follows

Agriculture is an important economic sector in Africa, including Ethiopia. The sector accounts for Africa’s 25% GDP, 21% exports, and 65 to 70% of the workforce supporting the livelihoods of 90% of the population [1-3]. In Ethiopia, agriculture contributes to 44% of the GDP, 70% of the export earnings and 80% of the employment opportunity [4].

Comment 2: In lines 201-204: please better explain how qualitative and quantitative tests were performed. What is t analysis? Do you mean the t-student test? What was the null and alternative hypothesis? What was explicitly tested? What is the significance level assumed?  This paragraph needs to be improved substantially with further information about the analysis for clarity.

Response 2: Thank you very much.  The data analysis section has been improved as follows:

The qualitative data collected were coded into a suitable category and captured with  quantitative data across the variables. Both data sets were subjected to data analysis using the Statistical Package for Social Sciences (SPSS) version 23 [46]. Descriptive statistics such as frequencies and percentages were computed using the cross-tabulation procedure. Significant tests were done with Chi-square test for qualitative and quantitative data sets. Contingency chi-square tests were employed to make statistical inference at the 0.05 level of significance to assess the relationship among variables. Conversely, the quantitative data for cost-benefit analysis were summarized using Microsoft excel to calculate the ratios.

Comment 3: Would you please add the significance level in all tables related to tests?

Response 3: Thank you. The significance level in all the Tables are included except for Table 10, which consisted of data collected through FGD where data analysis was done using the MS excel.

Comment 4: It does not make sense p-value = 0; this would mean complete certainty. For instance, in Table 2, in the chi-square test related to the number of children, the p-value should be 3.2 × 10-5, not zero. Please, review all tests performed in the manuscript, and fix p-value = 0 cases.

Response 4: Thank you for the comment. We have provided the output of the SPSS V 23 as it is.

Comment 5: Please, replace (p=) with (p-value=).

Response 5: Thank you. Revised.

Comment 6: Please, a table with the correlation between the quantitative descriptors collected in the manuscript, and report on results/discussion where best judged.

Response 6: Thank you for the esteemed suggestion. This study involves non-parametric socio-economic data. We strongly feel that the correlation analysis would not provide a fair comparison, discussion and conclusions. Hence, we have not included correlation analysis.

Comment 7: Please, provide a figure with a geographical heatmap of the cost-benefit ratio calculated across the geographical regions and discuss in the result section where appropriate.

Response 7: Thank you for this insight. Unfortunately, we do not have detailed data set across the various geographic regions to provide a heatmap. This is a valuable suggestion for our future research involving diverse agro-ecologies, socio-economists, agronomists and plant breeders.

Comment 8: Please, generate a principal component analysis plot displaying how the variables mentioned in Table 6 are closely or distantly related and discuss where best judged.

Response 8: Thank you. We have included a PCA plot, and the major features of this are discussed.

Comment 9: In lines 252-255: what are the favorable growing conditions? Would you please provide further information about the key drivers of the high yield?

Response 9: Thank you. The information is provided as follows

The use of different crop management methods such as weed management practices, crop rotation and row planting are the most favorable growing conditions and essential drivers for high yield gains. Furthermore, farmers in these districts had access to improved seed through the Bureau of Agriculture and research institutions which allowed higher yield gains. 

Comment 10: In lines 368-372: please discuss agricultural management practices used by the growers to justify maize as the least profitable crop.

Response 10: Thank you and discussed as follows:

Maize is one of the major crops in Ethiopia, including in the study areas. Nevertheless, farmers are not deriving profits from the production and marketing of this crop due to various reasons. The primary reason is that, in the country, the grain prices of maize are unpredictable due to the high market supply during the production season. This condition is the major constraint for maize farmers given that most of them have access to  the local markets to sell maize [74]. Also, there are no adequate postharvest infrastructures in the country, including transport, storage and processing.

Reviewer 2 Report

The research article is written nicely and the content are also good, its a laborious work to getting data from various farmers and to compile it in a frame. But concern is that the work presented in article to improve the genetic characteristics as well as overall improvement of finger millet to increase the production for generating more income in farmers community; is a kind of entension work; means raw data collected from the fields and from this data future work will be planned. In my opinion this is more suitable to entension work related Journals.   

Author Response

Thank you for the nice suggestion

Reviewer 3 Report

Interesting work suppoted by original data.

Author Response

Thank you

Reviewer 4 Report

Dear authors, 

The manuscripst is well writen but I have a few questions about it: 

What is the agricultural importance of Ethiopia for Africa?

Explain how qualitative and quantitative tests were performed.

What agricultural management practices were used by the growers to justify maize as the least profitable crop?

What are the favorable growing conditions? provide further information about the key drivers of the high yield.

Add the significance level in all tables related to tests.

Author Response

Comment 1: What is the agricultural importance of Ethiopia for Africa?

Response 1: Revised as follows

Agriculture is an important economic sector in Africa, including Ethiopia. The sector accounts for Africa’s 25% GDP, 21% exports, and 65 to 70% of the workforce supporting the livelihoods of 90% of the population [1-3]. In Ethiopia, agriculture contributes to 44% of the GDP, 70% of the export earnings and 80% of the employment opportunity [4].

Comment 2: Explain how qualitative and quantitative tests were performed.

Response 2: Thank you very much.  The data analysis section has been improved as follows. This  detail has been provided in the data analysis section:

The qualitative data collected were coded into a suitable category and captured with  quantitative data across the variables. Both data sets were subjected to data analysis using the Statistical Package for Social Sciences (SPSS) version 23 [46]. Descriptive statistics such as frequencies and percentages were computed using the cross-tabulation procedure. Significant tests were done with Chi-square test for qualitative and quantitative data sets. Contingency chi-square tests were employed to make statistical inference at the 0.05 level of significance to assess the relationship among variables. Conversely, the quantitative data for cost-benefit analysis were summarized using Microsoft excel to calculate the ratios.

Comment 3: What agricultural management practices were used by the growers to justify maize as the least profitable crop?

Response 3: Thank you and discussed as follows:

Maize is one of the major crops in Ethiopia, including in the study areas. Nevertheless, farmers are not deriving profits from the production and marketing of this crop due to various reasons. The primary reason is that, in the country, the grain prices of maize are unpredictable due to the high market supply during the production season. This condition is the major constraint for maize farmers given that  most of them have access to  the local markets to sell maize [74]. Also, there are no adequate postharvest infrastructures in the country, including transport, storage and processing.

Comment 4: What are the favorable growing conditions? provide further information about the key drivers of the high yield.

Response 4: Thank you. The information is provided as follows

The use  of different crop management methods such as weed management  practices, crop rotation and row planting are the most favorable growing conditions and essential drivers for high yield gains. Furthermore, farmers in these districts had access to improved seed through the Bureau of Agriculture and research institutions which allowed higher yield gains. 

Comment 5: Add the significance level in all tables related to tests.

Response 5: Thank you. The significance level in all the Tables are included except for Table 10, which consisted of data collected through FGD where data analysis was done using the MS excel.